# Hunting strategies to increase detection of chronic wasting disease in cervids

Atle Mysterud ⬤ [1✉], Petter Hopp ⬤ [2], Kristin Ruud Alvseike[3], Sylvie L. Benestad[2], Erlend B. Nilsen ⬤ [4], Christer M. Rolandsen ⬤ [4], Olav Strand[4], Jørn Våge[2] & Hildegunn Viljugrein ⬤ [1,2]

The successful mitigation of emerging wildlife diseases may involve controversial host culling. For livestock, 'preemptive host culling' is an accepted practice involving the removal of herds with known contact to infected populations. When applied to wildlife, this proactive approach comes in conflict with biodiversity conservation goals. Here, we present an alternative approach of 'proactive hunting surveillance' with the aim of early disease detection that simultaneously avoids undesirable population decline by targeting demographic groups with (1) a higher likelihood of being infected and (2) a lower reproductive value. We applied this harvesting principle to populations of reindeer to substantiate freedom of chronic wasting disease (CWD) infection. Proactive hunting surveillance reached 99% probability of freedom from infection (<4 reindeer infected) within 3–5 years, in comparison to ~10 years using ordinary harvest surveillance. However, implementation uncertainties linked to social issues appear challenging also with this kind of host culling.

[1] Centre for Ecological and Evolutionary Synthesis (CEES), Department of Biosciences, University of Oslo, P.O. Box 1066 Blindern, NO-0316 Oslo, Norway. [2] Norwegian Veterinary Institute, P.O. Box 750 Sentrum, NO-0106 Oslo, Norway. [3] Norwegian Food Safety Authority, Head Office, P.O. Box 383, N.2381 Brumunddal, Norway. [4] Norwegian Institute for Nature Research (NINA), P.O. Box 5685 Torgarden, NO-7485 Trondheim, Norway. ✉email: atle.mysterud@ibv.uio.no

Culling is often used to control wildlife disease outbreaks[1], but is invasive and often in conflict with other management objectives[2,3]. Early action is important to combat diseases that build up environmental reservoirs, such as anthrax[4] and chronic wasting disease (CWD)[5]. For diseases with latent stages or extended periods of low prevalence leading to low detectability during the early epidemic stages, mitigation may involve "preemptive culling" as a proactive measure by removing contact herds before the disease is detected[6]. Although the use of preemptive culling is widespread for livestock and beneficial from a disease mitigation perspective[7], this measure may not be a politically feasible option for wildlife species of conservation concern.

Here, we develop a general approach of "proactive hunting surveillance" aimed at detecting wildlife diseases at early epidemic stages as an alternative to preemptive culling. We apply empirically proactive hunting surveillance to a recent outbreak of CWD in wild reindeer (*Rangifer tarandus*) in Norway[8]. CWD is a fatal prion disease that affect cervids[9], currently spreading geographically in North America[10], causing population declines in endemic areas[11,12]. The Norwegian government aims for disease eradication to prevent a similar situation from developing in Europe, and the population with established CWD infection was eradicated[13]. However, the infection status of adjacent populations remains uncertain[14]. Reindeer and caribou face population declines across the northern hemisphere[15]. These populations are all part of the southern European conservation region for wild reindeer (Fig. 1). Preemptive culling is therefore politically difficult and not desirable from a conservation viewpoint.

CWD surveillance relies on testing hunter-killed cervids[16]. To increase the chance of detecting the disease in its early stages by harvesting, massive sampling is required[17,18], but it does not usually involve strategic plans for selective harvest. Unless well planned, this massive harvesting may be unsustainable. The targeted sampling of specific demographic groups may enhance the probability of disease detection[19]. CWD in cervids has a higher prevalence in adults than in yearlings and calves, and it is usually higher in adult males than in adult females[20–22]. For polygynous species, males are typically not limiting for population growth, unless the sex ratios are extreme[23], and increasing the harvest of specific demographic groups may be used to reduce impacts on population growth. The idea of creating a detailed plan for harvest quotas to selectively hunt for particular demographic groups to enhance disease detection but simultaneously avoid undesirable population declines is novel.

Here, we explore the general principle as to how this trade-off between sampling by increased and selective harvest and avoiding undesirable population declines may be solved. We determine, by aid of a simulation model, the optimal size and demographic composition of the harvest when the aim is to rapidly substantiate freedom from infection, without causing a notable population decline. Our approach hence merge traditional wildlife management principles of selective harvesting[24,25] with concepts of freedom from infection and risk-based surveillance coming from veterinary epidemiology[26,27]. This framework has direct applications to CWD in cervids, while the general approach can be extended to other infectious diseases.

## Results

An overview of the three model compartments is given in Fig. 2, and Supplementary Table 1 for details on parameters. We used stochastic simulations to determine how the optimal size and composition of harvest (given management aims) vary depending on the target sex ratio, the initial size of the population, the demographic pattern of the infection (relative risk), the probability of infection introduction, and the so-called "design prevalence"[28]. The design prevalence is the prevalence of the infection that managers have decided that the surveillance should be able to detect to document freedom from disease. The simulation model of population dynamics and harvesting ("Population simulation model" in Methods) was parametrized to the targeted reindeer populations (either Hardangervidda or Nordfjella zone 2) using demographic rates and initial population size, sex-, and age-structure estimated from four annual population surveys ("Population estimation model" in Methods). The output of the population simulation model (yearly number harvested, population size, sex-, and age-structure) was connected to a disease detection model to account for demographic pattern of infection and sensitivity of the test regime with the examined samples at hand ("Disease detection model" in Methods). The final output of the simulations from the disease detection model provided estimates of probability of freedom from infection for the specific design prevalence, for each scenario of harvest strategy and combination of epidemiological parameters ("Disease detection model").

**Demographic composition of harvest**. The ordinary harvest rate was 7.4% for calves, 7.7% for yearlings, 11.3% for adult females and 14.3% for adult males in Nordfjella zone 2 (Supplementary Table 2). If continuing with this rate and harvest composition, it would take 6 years (2018–2023) to yield 90% confidence in freedom from infection, 11 years (2018–2028) to reach 99% at the given prevalence of infection to be detected (design prevalence) and a relative CWD risk of 1:2:6 for yearlings:adult females:adult males (Fig. 3a). The optimal strategy was to cull all the available adult males during the first year and only shoot enough females to balance the population size within given levels (Fig. 3a). This naturally had marked impact on the demographic composition of the population, but allowed to retain a stable female population size (Fig. 4). When varying the sex ratio (m:f) threshold from 1:3, 1:10 to 1:20, it would take 3, 2, and 2 years to yield 90% confidence in the freedom from infection and at least 5, 4, and 3 years to reach 99%, respectively (Fig. 3a), for the given prevalence of infection to be detected (Supplementary Table 1). Calves and yearlings have low infection levels and add little information to detect the disease before they reach the adult stage. Nevertheless, retaining ordinary harvest rates of calves and yearlings only slightly decreased the estimated probability of freedom from infection after 5 years (Supplementary Table 3). In addition to the effect coming from relative risk among demographic groups, a proactive hunting surveillance compared to ordinary harvest has the added value of increasing sample size without causing a decline in the female population (Supplementary Table 4).

For Hardangervidda, the ordinary harvest rate was 13.9% for calves, 15.8% for yearlings, 14.3% for adult females and 18.0% for adult males (Supplementary Table 2). Continuing with this rate and harvest composition, it would take 4 years (2018–2021) and 10 years (2018–2027) to reach 90% and 99% confidence in freedom from infection, respectively. It would take 2, 1, and 1 years to yield 90% confidence in the freedom from infection and at least 5, 4, and 3 years to reach 99% if setting the sex ratio (m:f) to 1:3, 1:10 or 1:20, respectively (Fig. 3b), while causing no decline in the female population (Fig. 4b). The effect of varying relative risk, the prevalence of infection to be detected and probability of introduction was qualitatively similar to Nordfjella zone 2 (Supplementary Fig. 1, Supplementary Tables 5-6).

**Demographic infection pattern and relative risk**. For a more general inference, we explored the variation in the relative risk of 1:1:1 (no pattern) and 1:2:2 (e.g., assumed for CWD in elk *Cervus*

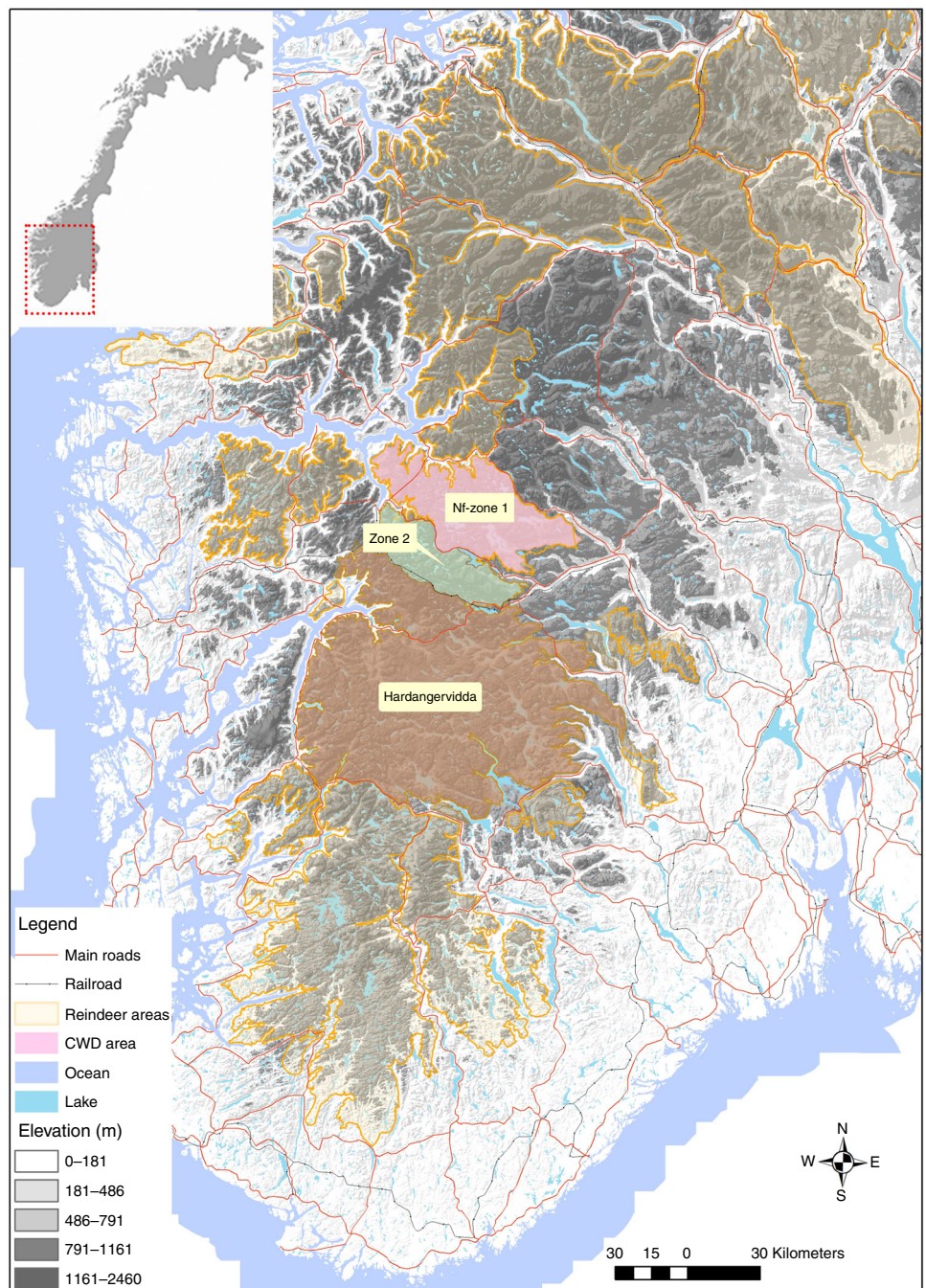

**Fig. 1 The location of the study populations.** The populations in Nordfjella zone 2 and Hardangervidda have connectivity to the earlier CWD-infected reindeer population of Nordfjella zone 1, Norway.

*canadensis*) relative to our baseline of 1:2:6 (e.g., CWD in white-tailed deer *Odocoileus virginianus*, mule deer *Odocoileus hemionus* and reindeer). The relative risk had a stronger influence on the disease detection when a higher proportion of adults were harvested, and in particular adult males, compared to other demographic groups (Fig. 5a). When the strategy was harvesting only adults and all available adult males to a threshold sex ratio (m:f) of 1:5, the estimated probability of reaching freedom from infection in 2018 increased from 70% for 1:1:1 to 73% for 1:2:2, and to 84% for 1:2:6 in Nordfjella zone 2 (Fig. 5a) and from 78% for 1:1:1 to 80% for 1:2:2 and to 90% for 1:2:6 for Hardangervidda (Supplementary Fig. 1). Upon surveilling an ordinary harvest (using the average hunting rates from the previous 3 years), the

patterns of harvesting in these regions were less biased towards males (Supplementary Table 2), lowering the difference between the outcomes of varying the relative risk.

**Design prevalence and management**. In reindeer, setting the epidemiological characteristics for CWD is primarily based on expert judgement (Supplementary Table 1) and given that the management decision is aimed at early detection. The time needed to reach a given level of confidence in freedom from infection was directly linked to the design prevalence, i.e., the prevalence of infection that surveillance should be able to detect (Fig. 5b). Detecting a level of 4 infected individuals in a small and a large

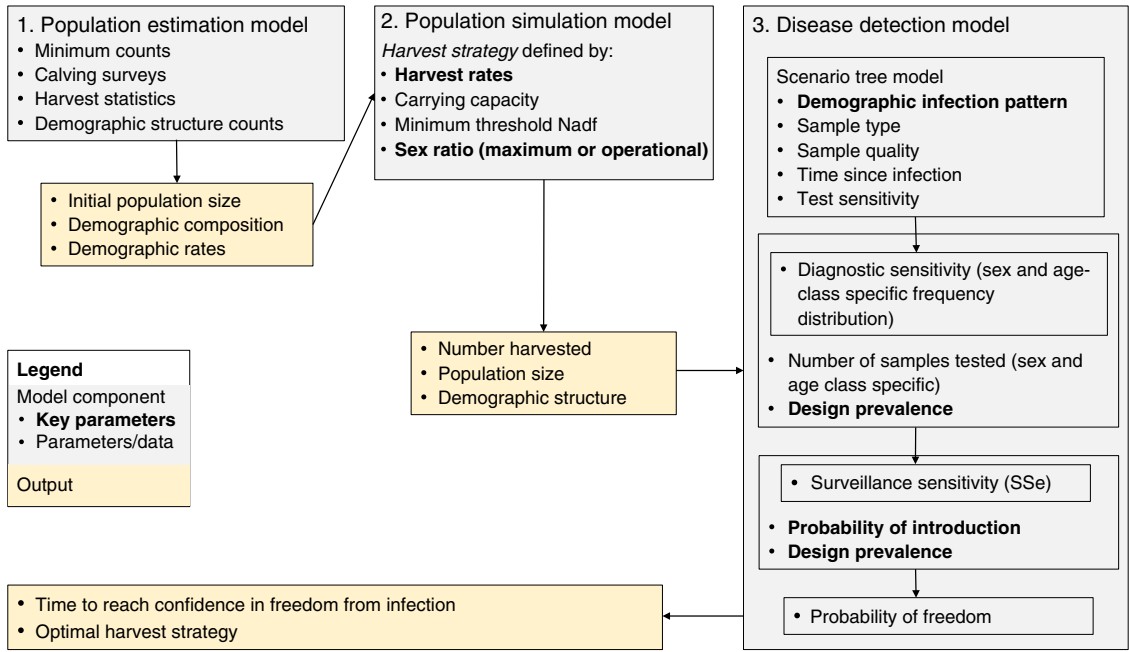

**Fig. 2 An overview of the modeling approach.** The model consists of three compartments and steps giving output to be used in the next model. The surveillance sensitivity (SSe) is the probability of detecting the disease from the specific sampling regime and at the specified design prevalence, the latter being the prevalence of infection to be detected. SSe is calculated for each year on the basis of the simulated data. Together with the design prevalence and risk of infection introduction from a year to another, the yearly SSe is used to update the estimated probability of freedom from CWD for each successive year. The model is run for various combinations of selected values of key parameters (marked in bold) and compared with respect to the time to reach confidence in freedom from infection (cfr. Supplementary Table 1).

population of 500 and 7500 (excluding calves), respectively, corresponds to a design prevalence of 0.008 for Nordfjella zone 2 and 0.00053 for Hardangervidda. Therefore, reaching a probability of 99% when detecting infection in 4 infected individuals in the population, would take approximately 10 years in both populations if testing 18% of the adult population each year, corresponding to yearly samples of 72 and 1080 test individuals from Nordfjella zone 2 and Hardangervidda, respectively. This scenario were estimated under the assumptions of a stable population of 20% yearlings and 80% adults, ignoring the different probabilities of infection between adult females and males (i.e. RR = 1:2:2), and assuming that both populations had reached 55% probability of freedom in 2017.

**Probability of introduction**. The probability of infection introduction is another expert judgement. After eliminating the CWD-infected population, this probability was low but not absent due to the ability of the prions to sustain their infectivity in the environment for years. The risk of introduction (Fig. 5b) delays the establishment of freedom from infection. With a 5% introduction risk (corresponding to 1 introduction per 20 year), a level of 99% will not be reached within 10 years at the current design prevalence with a sex ratio threshold of 1:3 or 1:5.

**The model meets reality**. For Nordfjella zone 2, the Norwegian Food Safety Authority requested higher harvesting rates prior to the 2018–2019 hunting seasons, to reach 90% freedom from infection with the given prevalence of infection to be detected (Supplementary Table 1). The model outlined here was used to suggest the required offtake as a basis for setting quotas that would meet the management target for early disease detection probability and simultaneously avoid population decline. In practice, the offtake was much lower as the whole quota was not

filled (Fig. 3). Only 63% (95% central range interval, CI: 62–65%) certainty regarding freedom from infection was reached. To mitigate risk, an extra culling of 50 adult males and 2 adult females was conducted winter 2019 by marksmen, which increased the certainty to 75% (73–77%) freedom from infection in Nordfjella zone 2. After new ordinary hunting in 2019, this level increased to 86% (95% CI: 84–88%) (Fig. 3a). Using a stochastic relative risk slightly increased uncertainty (95% CI: 83–88%). Ordinary hunting on Hardangervidda in 2018 led to 68% (67–69%) freedom from infection (depending somewhat on how to include missing data on the age category of hunted animals). Before hunting in 2019, a marked increase in the adult male harvest was planned based on our model principles, increasing the harvest to ~1170 males and 86% (95% CI: 84–87%) certainty of freedom from infection (Fig. 3b).

## Discussion

The geographic expansion of CWD to Europe was in 2018 rated among the most important new global issues for biodiversity and conservation research[29]. The issue of emerging diseases and culling as a mitigation tool is, however, not novel in Europe. Badger culling in the UK has long been highly controversial[3]. The CWD outbreak in cervids and ongoing surveillance in Europe[30] and the recent epidemic spread of African Swine Fever (ASF) among wild boar have brought disease questions to a whole new audience in the wildlife management of Europe. The planned culling of wild boar in Poland to combat ASF is met with considerable resistance among conservationists, and it is questioned by those with professional expertise[31]. The emergence of CWD highlights a dilemma in the discovery and surveillance of wildlife diseases when its diagnostics require the *post-mortem* testing of a species of conservation concern. Our model provides a basis for how to optimally manage wildlife populations when using culling

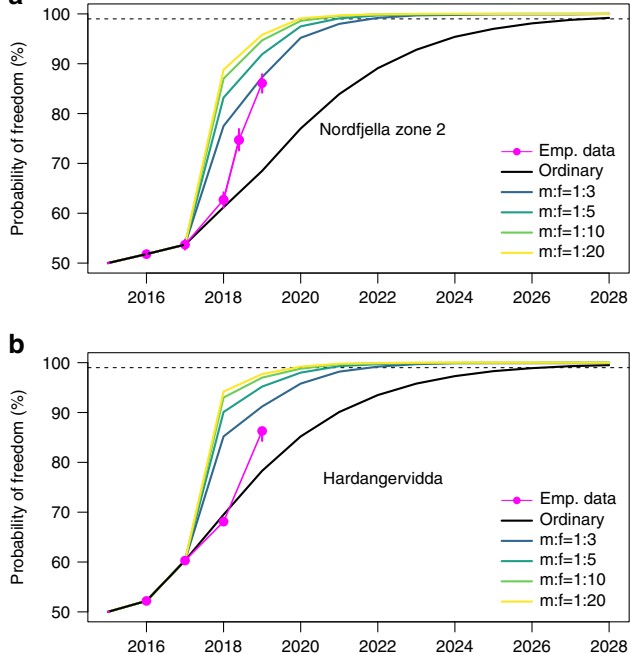

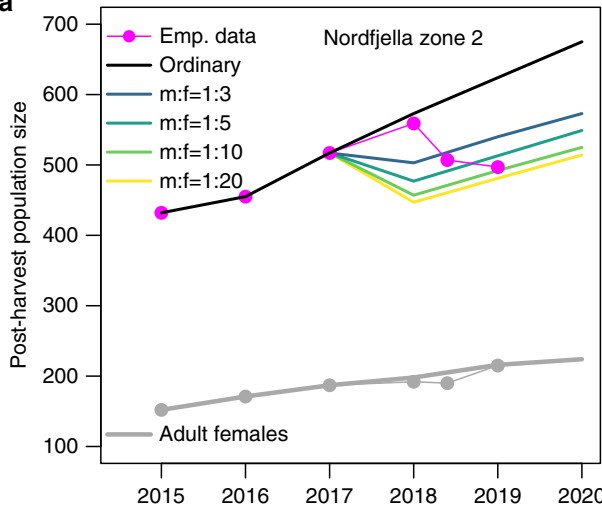

**Fig. 3 The effect of surveillance on time to freedom-of-infection.** The effect of the demographic harvest composition on the time to reach a given probability of reaching freedom from infection depending on the thresholds set for the adult sex ratio (male:female ratios). Curves describing the effect of the specific harvest strategy are given for each sex ratio relative to the normal sex and age-specific quotas found historically ("ordinary", black line) and the actual trajectory when trying to implement the principle of proactive hunting surveillance with a male biased harvest resulting in a female-biased sex ratio in the remaining population ("emp. data", magenta line with circles). (**a**) Nordfjella zone 2 (small population) and (**b**) Hardangervidda (large population). The error bars representing the 95% central range of 1000 simulations is included in both figures, but were too narrow to be visible before 2019 for Hardangervidda.

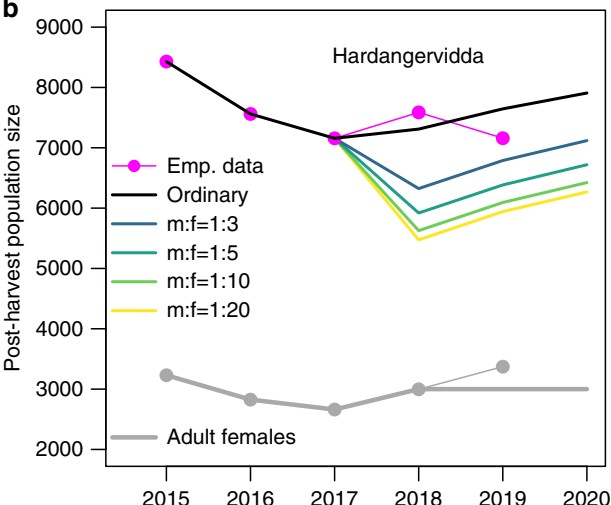

**Fig. 4 The effect of surveillance on the population.** The effect of proactive hunting surveillance on population size and demographic structure for the reindeer population in (**a**) Nordfjella zone 2 and (**b**) Hardangervidda. For the hunt in 2019, a quota was set based on an adult sex ratio of 1:5.

to increase disease detection and at the same time minimize undesirable population declines.

Demographic patterns of infection are common in wildlife[32,33], and related insight is useful for enhancing disease detection. Moreover, it is well recognized that targeted demographic offtake affects population dynamics. The males of polygynous species do not limit population growth under most circumstances[23], and empirical data suggest sex ratios must be extremely female-biased before impregnations are affected[34]. Setting a lower proportion of males needed for the impregnation of females has proven critical for modeling an optimal harvest[24]. With a three times as high probability of CWD infection in adult males compared to adult female cervids[22], the rapid establishment of freedom from infection implied that harvesting all the adult males during the first year reached the model threshold for the lowest proportion of males. Relative to the proportional harvest set historically by local management, engaging proactive hunting surveillance was able to decrease the time to 90% freedom from infection by 3–4 years and to 99% by 6–7 years (with sex ratio threshold of at least 1:5, Fig. 3), several years earlier compared to surveillance of the ordinary harvest. For the practical management situation, a modest 1:5 sex ratio was set for Hardangervidda to offset the potential for the undesirable side effects (see discussion below) of the resulting change in the demographic composition (Fig. 4). Depending on the disease risk situation, it is also possible to extend the period required to reach a given level, to obtain a lower risk of adverse effects.

CWD appears not to cause any clear sex-biased infection in elk[35–37], and there was no clear sex pattern in infection prevalence of white-tailed deer in one endemic area[11]. A lower variation in relative risk among the sexes would reduce the effect of proactive hunting surveillance (our scenario using the relative risk of 1:2:2 for yearlings:adult females:adult males), but nevertheless biasing the harvest towards males would create the possibility of retaining a larger population size. The demographic infection pattern may change during an epidemic. However, our aim was to detect disease in early epidemic stages. Early epidemic stages for CWD can easily be up to a decade long, and to model a constant demographic infection pattern appear sufficient for our setting. However, at an early stage, stochasticity may be important due to low number of infected individuals. When including relative risk of infection as a stochastic distribution, there was a corresponding slight increase in the credibility interval of the estimated freedom from infection. There is huge variability in the infection patterns of many infectious wildlife diseases[32,33]. Other diseases require different relative risks among demographic groups compared to CWD, and heterogeneity in infection within

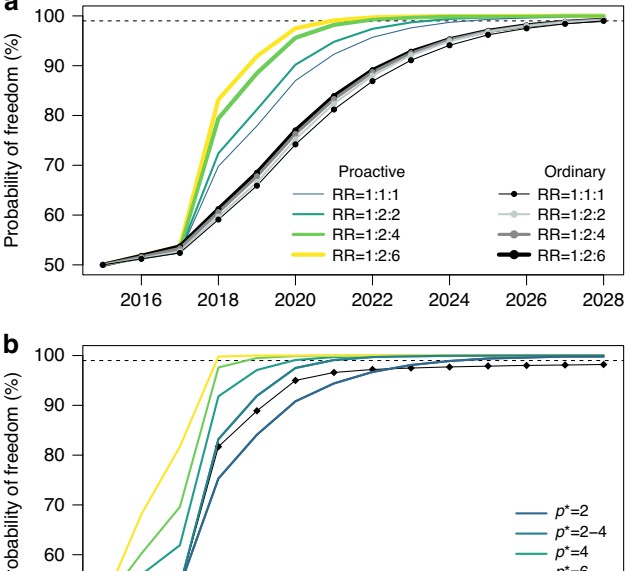

**Fig. 5 The effect of epidemiological characteristics on time to freedom-of-infection. a** The demographic pattern of infection is background for setting the relative risk (RR) of infection among age and sex classes. The effect of varying the RR is compared between a strategy of culling only adults and also culling available adult males to reach a sex ratio (m:f) of 1:5 and the strategy of surveillance for the ordinary harvest. **b** The effect of epidemiological uncertainty on the time to reach a given probability of freedom from infection. Design prevalence ($p^*$) is the prevalence of infection to be detected, which will depend on the assumed time of infection, rate of transmission and assessed importance of early detection. Risk of infection introduction (pIntro) were set to 5% and 0.1%, corresponding to before and after the adjacent infected population was culled, and compared to a scenario of keeping a high risk of introduction (5%) throughout all years. Due to a lack of relevant empirical data, these values were based on expert opinion.

demographic groups may lead to overestimates of confidence in freedom from infection[18]. Our case of CWD is complex including sophisticated models for both disease detection (to account for changing test sensitivity of different tissues during infection) and the population estimation (due to four annual surveys), but the general principle is simpler and provide a general tool potentially useful in a number of other settings. What is needed for proactive hunting surveillance to be useful is (1) a relative risk in the form of a clear infection pattern among classes of units, (2) selective harvesting is possible based on these units, and (3) a population model to predict one year ahead. In our case, we have a clear sex and age-specific pattern of CWD infection, and hunting of cervids is typically selective for these classes. Our model can easily be modified to explore a wider range of demographic infection patterns. In principle one may also include other factors leading to heterogeneous probability of infection, either temporal (seasonal infection) or spatial (distance to known infection) depending on disease system details.

Early detection and rapid action is required for the successful mitigation of many wildlife diseases[38]. In the case of CWD, environmental contamination with CWD prions become increasingly important for transmission during an epidemic outbreak[39], and the environmental persistence of prions makes disease mitigation difficult during the late epidemic stages[5]. A

critical issue when estimating the sample size is to set the level of infection one aims to detect, termed "design prevalence" in this context[28]. It is a management decision to set this target based on an understanding of a given epidemic situation and the evaluation of the risk. The common target of discovering a CWD prevalence of 1% in the USA was not sufficiently low in large populations of white-tailed deer if aiming for the rapid detection of a CWD outbreak[16], because it may correspond to a large absolute number of infected individuals. For this reason, the Norwegian Food Safety Authority used a design prevalence set as a number of individuals rather than as a proportion of the population. Their aim was to set the design prevalence sufficiently low to imply a high certainty regarding the absence of CWD and furthermore to require 99% confidence of the result to further reduce the risk that CWD would remain undetected. The design prevalence had a major impact on the time needed to reach a given level of freedom from infection.

The spatial structure of host populations is important for understanding and combating disease invasions[40]. The spatial targeting of sampling was important for CWD discovery among forest-dwelling cervids in North America with no clear population boundaries[41–44]. The situation with CWD in alpine reindeer is distinct from the situation in North America with forest-dwelling cervids in open populations, such as mule deer, white-tailed deer and elk[45]. Alpine reindeer live nomadically within more discrete populations, typically with limited connectivity to neighboring populations[46]. It is therefore possible to use some of the two level sampling design for culling, as typically used to combat disease in livestock organized on farms[47]. The spatial configuration of populations relative to the previously CWD-infected population affect the probability of introduction by host movement. We therefore used a lower probability of introduction for the more distant Hardangervidda population than in Nordfjella zone 2, which is bordering the previously CWD-infected and now eradicated population (Fig. 1). For our purpose, a fixed and quite high introduction probability was set during 2016–2017 before eradication was completed, while it was low after the CWD-infected population was eradicated. European landscapes are in many places highly fragmented[48]. Detailed knowledge of migration routes and barriers enabling the demarcation of functional populations would be important before implementing similar approaches to the more open populations typical of red deer (*Cervus elaphus*), moose (*Alces alces*) and roe deer (*Capreolus capreolus*).

The selective harvest of male ungulates may have unintended demographic side effects by changing their rutting behavior[23]. Removing a high proportion of males led to later and less synchronous rutting in semi-domestic reindeer[49], and even more for the less polygynous moose[50]. These side effects are usually not very strong, and we consider that these adverse effects can be offset against the chance of detecting the disease early. Evidence is building up to show that females perform an active mate search in ungulates[51,52]. It remains uncertain whether the absence of prime-aged males may cause herds of female reindeer to move out of the area in search of more preferable mates, which would represent a risk of disease spread. Therefore, from a precautionary principle standpoint, we recommend that the sex ratios should not be too extremely female-biased, in particular for less polygynous species.

Harvesting calves gives no information regarding the disease status in the case of CWD. However, shooting female adults without an offspring harvest may lead to orphaning with adverse effects on their growth and survival[53]. Hunters generally have a strong reluctance to harvest females with offspring[54]. The ethical aspects of shooting mothers without shooting dependent offspring are likely to come into play when confronting managers.

Our simulations showed that continued harvest of some calves did not markedly affect the time to freedom from infection, and retaining some calf harvest may reduce the number of orphans.

Implementation uncertainty may arise from both environmental and the human dimension. To account for environmental variation, our population model was stochastic for annual variation in survival and recruitment. It is more difficult to predict the social responses and level of quota filling. Due to such variation, we nevertheless used and will recommend annual updates of numbers following adaptive management protocols. Proactive hunting surveillance provides a platform of following scientific principles on how to harvest a cervid population optimally, with management aims linked to limiting emerging wildlife diseases. However, these principles counter the harvest composition of the current hunting traditions in Europe[55]. The management aims for the excessive harvest of white-tailed deer to combat CWD in Wisconsin, USA[2] and recently for wild boar to combat ASF in Poland[31] are notoriously unpopular and often fail to reach their harvest targets[3]. Such implementation uncertainty can be lowered by using marksmen. The general public may prefer professionals with lower wounding rates[56], as they are often more effective and trained at maintaining high standards of disease containment. The sharpshooter program in Illinois has been successful in limiting CWD[57,58], but it has led to a continued high level of conflict[59]. In Norway, marksmen were used for additional culling in Nordfjella, but similarly to the USA, these actions cause conflicts with local stakeholders[60]. To design hunting regimes that could be implemented practically while increasing the harvest offtake, local stakeholders were involved with epidemiologists at the Norwegian Veterinary Institute who were more systematically explaining data and logic of the model, which has been termed participatory modeling[61]. These efforts proved successful to lower conflicts and the local reindeer tribunal used the information to set a quota based on our model for proactive hunting surveillance. However, considerable controversy among laymen nevertheless arose when the quotas were known to the general public. Changing the attitudes of hunters and stakeholders will be challenging and controversies are likely to persist when implementing alternative harvesting regimes.

## Material and methods

**Study areas**. The study populations are situated in the southern mountain ranges of Norway (Fig. 1). Due to fragmentation, alpine reindeer in Norway are managed in 24 different wild reindeer areas, and ongoing fragmentation further limits movements within these reindeer management areas to the Nordfjella region. The Nordfjella reindeer region is managed independently in two separate areas termed zone 1 and zone 2. The Nordfjella zone 1 measures ~2000 km² and currently remains without reindeer since the CWD-infected population was removed[13]. The adjacent population, Nordfjella zone 2, measures ~1000 km², and it hosts the most closely connected population of wild reindeer southwest of the infected part of Nordfjella zone 1. The Hardangervidda mountain plateau covers 8000 km², and it contains by far the largest wild alpine reindeer population in Norway.

Reindeer populations are regulated by hunting[62]. For each reindeer management area, sex and age-specific quotas are set. The ordinary hunting season runs between the 20th of August and the 30th of September. This season was extended from the 10th of August, 2018 and the 1st of August, 2019 in the Nordfjella zone 2 and from the 10th of August, 2019 on Hardangervidda to aid in reaching the targets of the proactive hunting surveillance. Additional culling (50 adult males and 2 adult females) was performed by marksmen from the Norwegian Environmental

Agency (unit SNO) in the Nordfjella zone 2 population from January–April of 2019.

**Model components**. We here give some further details on the simulation model of population dynamics and harvesting (see "Population simulation model" below). This population simulation model is parametrized using demographic rates and initial population size, sex- and age-structure estimated from fitting the population estimation model to four annual surveys (see "Population estimation model" below). The output of the simulation model (yearly numbers of harvested, population size, sex and age structure) was coupled to a disease detection model to account for demographic pattern of infection and given sensitivity of the test regime with the samples at hand (see "Disease detection model" below). See Fig. 2 for an overview of model components and output, and Supplementary Table 1 for details of parameters.

**Population estimation model**. Estimations of population size and demographic composition into broad age classes were based on four different annual surveys of the reindeer populations in Norway[62,63].

(1) Minimum counts were performed during the winter using aerial surveys. These aerial surveys provide total counts irrespective of sex or age.

(2) Calving surveys are performed mid-summer (primarily the first half of July, when the calves are approximately 6–8 weeks old), also using airplanes. These surveys distinguished calves from yearlings and adult females. Only female herds are surveyed, but there is an unknown proportion of yearling males in these herds.

(3) Harvest data is highly reliable in Norway, and the numbers of calves, yearlings and adults of each sex are recorded.

(4) Demographic structures (sex and age composition) are reliably counted from the ground after the hunt and during the rut, because the populations are not sexually segregated during this period.

Based on the census data, we estimated reindeer abundance, population age- and sex-structures as well as demographic rates (i.e. survival probabilities and reproductive rates) using a hierarchical change-in-ratio model with parameters estimated using Bayesian inference described in Nilsen and Strand[63]. In this model, information about changes in population structures following a removal of a known number of reindeer with known age- and sex distribution by harvest was utilized to estimate the parameters of interest. Estimating parameters based on Bayesian inference allowed us to exploit the great flexibility of the BUGS language, facilitating the use of data from a sampling scheme that does not match the assumptions of traditional estimators. As a core in the process model, we used a stage-dependent two-sex matrix model with three age classes: calves, yearlings and adults. We allowed recruitment rates and calf summer survival to vary stochastically among years, whereas a constant winter survival was assumed (common for each sex and age class).

For the basic formula (in jags), see model variant 2 in the r-script "M1_variant_Hardangervidda_fall2018.R" at the separate github-link (https://github.com/ErlendNilsen/CiR_usage). Priors were specified as uninformative, using uniform distributions (0-1) for probabilities and very wide normal distributions for initial population sizes. The model was run in R through rjags using three chains of 250,000 iterations, a burn-in period of 50,000, and the chains were thinned by 3. Convergence was assessed by visual inspection of MCMC-chains and the Gelman–Rubin statistics. Output from the model for year 2018, is shown in Supplementary table 2 and was used as input for simulations from the population model.

**Population simulation model**. We use the population simulation model as a basis to simulate the population data for years to come, using the demographic rates and initial population size and structure as estimated from the population estimation model based on the data from our two study areas. The core process is a stage-dependent two-sex matrix model with three age classes, namely, calves, yearlings, and adults, where the transition from a year to the next is dependent on harvest, survival and recruitment rates. We used 1000 iterations for each scenario. Stochastic input parameters (demographic rates and initial population size of each sex and age class) were randomly drawn from probability distributions as specified by mean and standard deviation (sd) (Supplementary Table 2). The population parameters involve both environmental constraints (carrying capacity), thresholds to limit undesirable population declines (minimum female population size), thresholds for the sex ratio and the harvest we aim to optimize (overview in Supplementary Table 1):

- *Operative sex ratio*. We explored the adult male to female ratios from 1:3, 1:5, 1:10 and up to 1:20, because it will cover the primary range of cervid mating systems. We set these ratios as thresholds to maintain after hunting each year, i.e., in the population entering the rutting season.
- *Minimum number of adult females*. We set a minimum number of adult females after the harvest as a threshold. Note that the threshold for females will markedly affect the number of males if managed by a certain sex ratio.
- *Carrying capacity* ($K$). The populations would grow fast and overshoot $K$ if we were not shooting females. We therefore aimed to harvest a sufficient number of females to stabilize the population well below the $K$, which is set on the basis of the current management aim to avoid forage deterioration[62].
- *Harvest rate and composition*. By comparing simulations from different harvest strategies, this is the target we aim to optimize to inform management decisions. The harvest rates were set separately for males and females and the age categories of calves, yearlings and adults. A normal harvest strategy was set as the demographic composition and level of harvest averaged over three years (2016-2018). Alternative harvest strategies were specified according to reach a certain post-harvest operational sex ratio. For both types of harvest strategies ("ordinary" and "proactive"), the harvest rate of females varied according to $K$ (Supplementary Table 1), and it was constrained according to the threshold of minimum number of adult females.

From initial population size and structure, demographic rates, harvest rates and specific harvest strategies, population data and number of harvested were simulated for years to come. In the following, let c = calves; y = yearlings; ad=adults; m = males; f = females. For calves and yearlings, the harvest numbers (H) were derived from the specified harvest rates (h) (and rounded to nearest integer) for each year $t$:

$H_{cf}[t] = N_{cf}[t] \cdot h_{cf}$,
$H_{cm}[t] = N_{cm}[t] \cdot h_{cm}$,
$H_{yf}[t] = N_{yf}[t] \cdot h_{yf}$,
$H_{ym}[t] = N_{ym}[t] \cdot h_{ym}$.

In order to stabilize the population size well below the carrying capacity ($K$), harvest rate of adult females ($h_{adf}$) was for each year $t$ determined by the total population size ($Ntot[t]$) compared to $K$: $H_{adf}[t] = hscale - hscale \cdot (K - Ntot[t])/K$. The parameter *hscale* is shortening of the longer formula; $Hadf = hadf.max - hadf.m \cdot (K-Ntot[t])/K$, where $hscale = hadf.max = hadf.m$). *hscale* was tuned to make first year harvest rate of adult females to be close to the "ordinary" harvest rate (defined by the 3 years mean; see Supplementary table 1).

Number of adult females harvested ($H_{adf}[t]$) in year $t$ is set to the product of preharvest number of adult females times the harvest rate: $H_{adf}[t] = N_{adf}[t] \cdot h_{adf}[t]$. In addition, $H_{adf}[t]$ was rounded to nearest integer and was restricted so that: $N_{adf}[t] - H_{adf}[t] > T\_adf$. $T\_adf$ is the lower threshold of number of adult females after harvest.

We used two main types of harvest strategy for adult males:

Harvest strategy 1 "ordinary" (with threshold of maximum sex ratio): This is the ordinary (or 2·ordinary) harvest rate, i.e. set equal to historical levels. Number of adult males harvested ($H_{adm}[t]$) in year t is set to the product of preharvest number of adult males times the specified harvest rate ($h_{adm}$):

$H_{adm}[t] = N_{adm}[t] \cdot h_{adm}$. In addition, $H_{adm}[t]$ was rounded to nearest integer and restricted so that: $N_{adm}[t] - H_{adm}[t] > T\_adm[t]$. $T\_adm[t]$ is the lower threshold of number of adult males after harvest. This threshold is defined by the operational sex ratio; $SR = m:f = 1:SRadf$. The lower threshold of adult males are then determined by $SRadf$ and the number of post-harvest number of adult females: $T\_adm[t] = (N_{adf}[t]-H_{adf}[t])/SRadf$ (e.g., if $SRadf = 20$, at least 5% of post-harvest adults should be males).

Harvest strategy 2 "proactive" (with operational sex ratio): This is the harvest strategy to maximize disease detection within constraints given in particular by skew in sex ratio (Supplementary Table 1). Number of adult males harvested ($H_{adm}[t]$) in year $t$ is set to obtain a specified operational sex ratio ($SR = m:f = 1: SRadf$) after harvest. The aim for number of adult males after harvest are then determined by $SRadf$ and the number of post-harvest number of adult females:

$N_{adm}[t] - H_{adm}[t] = (N_{adf}[t]-H_{adf}[t])/SRadf$ (e.g., if $SRadf = 10$, at least 10% of post-harvest adults should be males).

Similar to the process model in the "Population estimation model" above, we used a binomial distribution to model the survival from one year to another, and to model the yearly reproduction of calves ($f[t]$) together with yearly calf summer survival ($phi1[t]$). We assumed that the winter survival ($phi3$) was the same across sex- and age classes. In the population simulation model, the demographic rates were modeled by gamma distributions, where the alpha and beta parameters of the gamma-distribution were derived as a function of the mean and standard deviation of the demographic rates (see Supplementary Table 2). Obtaining preharvest population sizes one year ahead:

$N_{yf}[t+1] \sim Binom(N_{cf}[t]-H_{cf}[t], phi3)$,
$N_{ym}[t+1] \sim Binom(N_{cm}[t]-H_{cm}[t], phi3)$,
$N_{adf}[t+1] \sim Binom(N_{yf}[t]-H_{yf}[t]+N_{adf}[t]-H_{adf}[t], phi3)$,
$N_{adm}[t+1] \sim Binom(N_{ym}[t]-H_{ym}[t]+N_{adm}[t]-H_{adm}[t], phi3)$,
$N_{cf}[t+1] \sim Binom(N_{adf}[t+1], phi1[t] \cdot f[t]/2)$,
$N_{cm}[t+1] \sim Binom(N_{adf}[t+1], phi1[t] \cdot f[t]/2)$.

**Disease detection model**. The primary principle of the disease detection model is presented in Viljugrein et al.[64]. This stochastic scenario tree model estimates the likelihood of detecting CWD infection in a given individual depending on its age class, our knowledge about infection development, and the sensitivity of the given testing regime. Here, we extend the previous model to also account for the sex-specific pattern of infection by including this as a risk factor. The expected duration of time from infection to death vary, but for the model we used a constant of 2 years for this purpose. The disease detection model relies on an understanding of the ability of the ELISA method to detect abnormal prion protein ($PrP^{Sc}$) during the course of CWD infection from samples of retropharyngeal lymph nodes (RLN) and brain tissue provided by hunters. Test sensitivity is increasing as a non-linear function of time since infection. In the simulations, infected

individuals are assumed infected at a random time from 0.5 week to 48 months or 38 months back in time, for adults and yearlings, respectively. We ignored all calves tested (because of the long time period before $PrP^{CWD}$ is found in the lymph nodes and the resulting low test sensitivity of calves) and assumed that male adults had a three times higher risk of being infected compared to adult females, which again had a double risk of being infected compared to yearlings.

In the simulations (1000 iterations), we randomly drew the time since infection for each of the tested individuals, and the resulting diagnostic sensitivity for individual $i$ in the simulation $j$ ($dSe_{ij}$) was determined by the pathway of individual $i$ through the scenario tree and weighted by the adjusted relative risk of the sex and age class. The pathway of an individual through the tree was randomly drawn according to probabilities specified by the testing regime.

The disease detection model was coupled to the estimated (2016–2018) and simulated (2019–2029) yearly population size and sex- and age-class structure, as well as the actual tested (2016–2019) and simulated reindeer being shot and tested each year (2018–2029). From simulations of harvesting for the years to come ("Population simulation model"), we assumed that all the reindeer being shot were also sampled and tested. We used 1000 iterations for each scenario of harvest strategy and combination of epidemiological parameters, and output results from the population model were used as input data to the disease detection model (Fig. 2). Here we present additional details using the following epidemiological parameters and model output (overview in Supplementary Table 1).

Epidemiological parameters:

- *Demographic infection pattern and relative risk.* The relative infection likelihood is defined based on the demographic infection pattern of CWD in reindeer[65]. We used a relative infection likelihood of 1:2:6 for yearlings:adult females:adult males unless otherwise stated. Calves were ignored due to the very low test sensitivity given their maximum time since infection was around 4 months (age at the time of hunting). We regard any further detail on the demographic pattern of CWD infection (i.e., slight increases with age among adult males) as not required for the current purpose. To explore how important the demographic pattern of infection is in time to reach a given likelihood of freedom from infection, we explored variants of 1:1:1, 1:2:2 and 1:2:4 for yearling to adult females and adult males. In the model, the relative risks are adjusted according to the demographic structure of the population to ensure that the average adjusted risk for a representative sample of the reference population (ignoring the calves) is 1, while the relative ratios were maintained as specified by Martin et al.[27].

- *Design prevalence.* In reality, freedom from diseases of an infectious agent cannot be proven. Therefore, documentation of freedom from infection usually means that prevalence of the infection, if present, is below a predefined level, the so-called design prevalence[28]. The design prevalence can be set either as a percentage or as a number of infected individuals relative to the specific population size. For specifying the design prevalence, we have only included the population of adults and yearlings, due to the low test sensitivity of calves. The CWD primarily has a frequency-dependent transmission rate, and it makes sense to set this as a number rather than as a percentage[16]. The resulting design prevalence (in %) would then be lower in a large population compared to a small one. We then varied the design prevalences to simulate the effect of varying the time since the introduction of disease, or to account for the uncertainty in the expected transmission rate of the disease.

- *Probability of introduction.* For the probability of introduction of infection, we used two scenarios partly based on expert opinion: 1) The probability of CWD introduction was set at 1% for Hardangervidda and 5% annually for Nordfjella zone 2 to reflect how distant/connected the population is from the source population (Nordfjella zone 1), respectively, and 2) the probability of introduction after eradication of the source population (Nordfjella zone 1) was reduced to 0.1% annually (i.e. 1 introduction per 1000 year). The probability of introduction is clearly higher with a CWD-infected population living nearby. We therefore also explored the theoretical impact of not eradicating the infected population in zone 1 in time to a given level of freedom from infection, i.e., by retaining a high probability of introduction.

The simulation model is stochastic and model output will be in the form of frequency distributions (summarized by mean and the 95% central range interval as defined by the 2.5th and 97.5th percentiles). The model output are:

- *Surveillance sensitivity.* The disease detection model calculates the annual surveillance sensitivity, the probability of detecting the infection (at least one sample testing positive) from the specific sampling regime and at the specified design prevalence[27]. A probability formula assuming infected reindeer are distributed according to the hypergeometric distribution, is used to calculate the surveillance sensitivity. Let $ProbAllSampleNeg_j$ denote the probability that there were no test-positive animals found in iteration $j$ of n samples for a given design prevalence, $p*$. According to MacDiarmid[66], a hypergeometric distribution can be approximated by: $ProbAllSampleNeg_j = (1 - \sum_{i=1}^{n} dSe_{ij} / PopSize_j)^{\wedge}(p* \cdot PopSize_j)$. $dSe_{ij}$ is the diagnostic sensitivity of individual $i$ in iteration $j$ (see above), $PopSize_j$ denotes the population size (ignoring calves) in iteration $j$ and $p*$ represents the design prevalence. For each of 1000 iterations, we then calculated the surveillance sensitivity as $1- ProbAllSampleNeg_j$.

- *Freedom from infection*: We expand on our previous disease detection model[64] here by including estimation of the probability of freedom from infection, i.e. that the prevalence of the infection is not higher than the design prevalence. To estimate the probability of freedom from infection, an updated posterior probability of freedom after each year of testing is calculated using the Bayes theorem, as described by Martin et al[27]. Assuming perfect specificity, the calculation of probability of freedom (PFree) is based on the prior probability of population being infected, priorPInf, and the surveillance sensitivity, SSe: $PFree = (1-priorPInf)/(1-priorPInf \cdot SSe)$. Further on, the calculation of priorPInf is based on the probability of infection introduction, pIntro, and the posterior probability of infection from the previous year, postPInf_tl1: $priorPinf = postPinf\_tl1 + pIntro - (postPInf\_tl1 \cdot pIntro)$, where postPInf_tl1 = 1 – prior probability of freedom (the probability of freedom estimated for the previous year). The prior probability of infection was set to 0.5 for the first year. This is a conservative estimate, and it corresponds to the lack of prior information about the infection status.

**Reporting summary**. Further information on research design is available in the Nature Research Reporting Summary linked to this article.

## Data availability

All data are available in a public repository[67].

## Code availability

All code is available in a public repository[67].

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

## Acknowledgements

The authors are grateful to Harald Skjerdal for the access to census data for the Nordfjella area and to Svein Erik Lund for discussions related to the census and modeling data for the Hardangervidda area. We are grateful to all the hunters and people involved in collecting and processing samples and to the Norwegian Veterinary Institute for the skilled and efficient analyses of the samples for CWD detection. The study was financed by the Norwegian Veterinary Institute project nr. 12081 (funded by the Norwegian Ministry of Agriculture and Food) and the Norwegian Environment Agency.

## Author contributions

A.M. and H.V. initiated the project. A.M. had the main conceptual idea and drafted the manuscript with major contributions to the methods and results parts provided by H.V. H.V. developed the disease detection model to include also freedom-of-disease together with P.H., and to utilize the reindeer population model developed by E.B.N. H.V. did simulations and made figures. K.R.A., C.M.R., J.V., and P.H. organized the CWD data collection and database management. S.L.B. is in charge of CWD testing. K.R.A. suggested increasing harvest to increase disease detection. O.S. organized the reindeer surveillance data collection. All authors edited and contributed to the drafts and gave final approval for publication.

## Competing interests

The authors declare no competing interests.
