## [Peer Review File · Nature Communications]

Reviewers' Comments:

Reviewer #1:

Remarks to the Author:

Mysterud et al. tackle the issue of how to manage and reduce the risk of a potential disease outbreak in a population or species that is of conservation concern (e.g., threatened). They develop and test a proactive hunting regime to: 1) survey for early detection of a disease and identify freedom from infection; while also 2) avoiding unwanted population decline. The authors use a combination of strong empirical data on population and disease parameters in concert with simulation to examine the effect of different hypothetical (and actual) management options on the probability that a population is free from infection. They develop and test their regime using the Chronic Wasting Disease outbreak in reindeer in Norway. There are many interesting results in this paper, but of most importance is that with proactive hunting measures (e.g., harvesting sex and age classes with the highest pattern of infection [or relative risk]), the authors demonstrate that a 99% probability of freedom from infection can occur in 1/3 of the time relative to ordinary harvest surveillance.

I find this article to be a fascinating example where ecological knowledge, theory, and modeling can be utilized to proactively manage and conserve a population or species. The pitch of the paper, empirical data, and modeling and simulation techniques are all strong. I only have a couple of comments:

- 1) The authors do not include how the environment influences the population. I am curious why not, and am curious if there is any reason that the environment could change significantly the predicted results and interpretation of the conceptualized proactive harvest regimes? Note that I do see that the authors include stochasticity in some of the demographic parameters, which does, at some level, take into account variation due to environment.
- 2) I found it pretty hard to read the results without a detailed read of Table 1 and/or reading the methods. For example, some of the terms are jargon and aren't defined until the methods (or in table 1). Design prevalence is one example. Including a bit more information/details when transitioning from the introduction to the results would be warranted (especially for the terms). Another option is to potentially provide more details in the figure legends, which would help them stand alone better and provide some more detail without having to skip ahead to the methods.
- 3) Some of the parameters are developed using partial expert opinion (e.g., probability of introduction). While I am confident the authors know their system well and discussed these parameters at length in developing the models, I wonder why simple sensitivity analyses were not done? Such an analysis can help 'verify' that results are not sensitive to certain choices of values.

Reviewer #2:

Remarks to the Author:

Review of Mysterud et al. "Hunting wildlife for disease detection"

The authors develop a surveillance framework for CWD using hunter harvest data. Hunter harvest is a common method of conducting disease surveillance but the authors suggest that recommendations for hunter harvest policies could be changed to optimize both population management and disease surveillance. Usually, hunting policies are only optimized for population management and disease surveillance is an opportunistic add-on. This aim is novel, interesting, and very worthwhile to pursue.

My main concern is that the framework doesn't consider disease dynamics. It relies on pre-existing knowledge of relative risk in different age classes. However, the interaction of demographic dynamics and R_0 / R_e /FOI will drive age-based infection risk (to change under different combinations of these rates) so I don't see how a constant assumption about age-based risk is valid.

I also found it difficult to understand all the components of the framework because much of it was developed as part of previous work, and they weren't combined in the Supp info with data input/model structure details so the reader has to go back to individual papers (although I did appreciate Fig. 2 as a broad overview) to understand how separate models were integrated. Relatedly, in its current form, I didn't feel the description of the framework is directly transferable to other systems, both in terms of minimally sufficient data to use the method and in terms of recommendations. I think the paper could benefit greatly from some kind of 'recommendations synthesis' maybe in the form of a decision tree that would help guide when different strategies would be used and which data would be required to use them.

L185 – provides

Table 1 – why is 'Principle and Basis' cell for 'Harvest rate and composition' blank?

Fig. 2 caption – what do the different colored text mean?

Fig. 3 caption – I'm not sure what the 'tested' trajectory is representing; what is the skew?

Fig. 4 – I don't find this figure very informative as it's completely predictable from the harvest strategies. It would be better to predict population abundance or growth rates as a metric of impact of the different strategies

SUPP info – L67 – it would be useful to provide some description of the type of model from Nilsen and Strand so one doesn't have to go looking, i.e., Bayesian hierarchical is super vague so the reader gets no insight as to which data are needed and how they are used.

SUPP info – L80 – same as above. There is not enough info here to understand what kind of model this is and which data are needed to use it.

Reviewer #1 (Remarks to the Author):

Mysterud et al. tackle the issue of how to manage and reduce the risk of a potential disease outbreak in a population or species that is of conservation concern (e.g., threatened). They develop and test a proactive hunting regime to: 1) survey for early detection of a disease and identify freedom from infection; while also 2) avoiding unwanted population decline. The authors use a combination of strong empirical data on population and disease parameters in concert with simulation to examine the effect of different hypothetical (and actual) management options on the probability that a population is free from infection. They develop and test their regime using the Chronic Wasting Disease outbreak in reindeer in Norway. There are many interesting results in this paper, but of most importance is that with proactive hunting measures (e.g., harvesting sex and age classes with the highest pattern of infection [or relative risk]), the authors demonstrate that a 99% probability of freedom from infection can occur in 1/3 of the time relative to ordinary harvest surveillance.

Our response: We are grateful for these positive comments!

I find this article to be a fascinating example where ecological knowledge, theory, and modeling can be utilized to proactively manage and conserve a population or species. The pitch of the paper, empirical data, and modeling and simulation techniques are all strong. I only have a couple of comments:

1) The authors do not include how the environment influences the population. I am curious why not, and am curious if there is any reason that the environment could change significantly the predicted results and interpretation of the conceptualized proactive harvest regimes? Note that I do see that the authors include stochasticity in some of the demographic parameters, which does, at some level, take into account variation due to environment.

Our response: This is a good question. Note that we are proactive for a rather short period of time (a few years). The potential problem is then stochastic annual variation in environment affecting survival and recruitment, rather than longer-term trends. Our model is correctly stochastic for both survival and recruitment to account for this, hence, unpredictable environmental variation similar to previous years is accounted for. We have added to the discussion on implementation: "Implementation uncertainty may arise from both environmental and the human dimension. To account for environmental variation, our population model was stochastic for annual variation in survival and recruitment. Due to such variation, we nevertheless used and recommend annual updates of numbers following adaptive management protocols. It is more difficult to predict the social responses and level of quota filling."

2) I found it pretty hard to read the results without a detailed read of Table 1 and/or reading the methods. For example, some of the terms are jargon and aren't defined until the methods (or in table 1). Design prevalence is one example. Including a bit more information/details when transitioning from the introduction to the results would be warranted (especially for the terms). Another option is to potentially provide more details in the figure legends, which would help them stand alone better and provide some more detail without having to skip ahead to the methods.

Our response: We are aware of the complexity of our modelling approach and certainly agree design prevalence is a term within veterinary epidemiology and not generally known, and we therefore clearly define it at the start section of results and have now mainly said "the prevalence of infected individuals to be detected". We have tried to improve this by explaining terms better also in the figure captions, in particular the overview of the model in figure 2. We now write in caption to figure 2: "The model consists of three compartments and steps giving output to be used in the next model. The surveillance sensitivity (SSe) is the probability of detecting the disease from the specific sampling

regime and at the specified design prevalence, the latter being the level of infected individuals to be detected. SSE is calculated for each year on the basis of the simulated data. Together with the design prevalence and risk of infection introduction from a year to another, the yearly SSE is used to update the estimated probability of freedom from CWD for each successive year. The model is run for various combinations of selected values of key parameters (marked in red) and compared with respect to the time to reach confidence in freedom from infection (cfr. Table 1). “. and to figure 5A: “(A) The demographic pattern of infection is background for setting varying relative risk (RR) of infection among age and sex classes. The effect of varying the RR is compared between a strategy of culling only adults and also culling available adult males to reach a sex ratio (m:f) of 1:5 and the strategy of surveillance for the ordinary harvest.”

3) Some of the parameters are developed using partial expert opinion (e.g., probability of introduction). While I am confident the authors know their system well and discussed these parameters at length in developing the models, I wonder why simple sensitivity analyses were not done? Such an analysis can help ‘verify’ that results are not sensitive to certain choices of values.

Our response: This is a valid and good comment. We had already played a bit around with variation of these parameters, but were uncertain how much to present in order not to overwhelm the reader. We have now added to the supplementary information some sensitivity analysis of the parameters chosen by expert opinion; most importantly the probability of infection introduction. We have already figure 5 showing how variation in design prevalence affect results. We have also added stochasticity in the demographic infection pattern (the relative risk). These results are presented as extra lines in our output in supplementary material. We have also briefly discussed the issue of stochasticity of the demographic infection pattern in connection with response to ref #2 on inclusion of disease dynamics.

Reviewer #2 (Remarks to the Author):

Review of Mysterud et al. “Hunting wildlife for disease detection”

The authors develop a surveillance framework for CWD using hunter harvest data. Hunter harvest is a common method of conducting disease surveillance but the authors suggest that recommendations for hunter harvest policies could be changed to optimize both population management and disease surveillance. Usually, hunting policies are only optimized for population management and disease surveillance is an opportunistic add-on. This aim is novel, interesting, and very worthwhile to pursue.

Our response: We are grateful for this overall positive comments!

My main concern is that the framework doesn’t consider disease dynamics. It relies on pre-existing knowledge of relative risk in different age classes. However, the interaction of demographic dynamics and R_0 / R_e /FOI will drive age-based infection risk (to change under different combinations of these rates) so I don’t see how a constant assumption about age-based risk is valid.

Our response: This is a very good comment. We are certainly aware that many diseases, including CWD, are likely to have different age-based infection risk depending on stage of the epidemic. However, since our approach is aiming for early detection, we can based on a series of papers (referred to in the paper) with good confidence assume a static age-based risk at low disease prevalence. To make this clear, we have added to the discussion: “The demographic infection pattern may change during an epidemic. However, our aim was to detect disease in early epidemic stages. Early epidemic stages for CWD can easily be up to a decade long, and to model a constant

demographic infection pattern appear sufficient for our setting. [...] Our model can easily be modified to explore a wider range of demographic infection patterns”.

However, we agree that stochasticity in the demographic infection pattern may be an issue in early stages. We have therefore added stochasticity in the relative risk to account for this uncertainty.

I also found it difficult to understand all the components of the framework because much of it was developed as part of previous work, and they weren't combined in the Supp info with data input/model structure details so the reader has to go back to individual papers (although I did appreciate Fig. 2 as a broad overview) to understand how separate models were integrated. Relatedly, in its current form, I didn't feel the description of the framework is directly transferable to other systems, both in terms of minimally sufficient data to use the method and in terms of recommendations. I think the paper could benefit greatly from some kind of 'recommendations synthesis' maybe in the form of a decision tree that would help guide when different strategies would be used and which data would be required to use them.

Our response: This was also a good point. We agree our example is fairly complicated in terms of detail. We have therefore given a synthesis recommendation for what is needed for a particular disease system to benefit from proactive hunting surveillance. We have expanded the part where we discussed how it can be used with different pattern of demographic infection: "Our case of CWD is complex including sophisticated models for both disease detection (to account for changing test sensitivity of different tissues during infection) and the population estimation (due to four annual surveys), but the general principle is simpler and provide a general tool potentially useful in a number of other settings. What is needed for proactive hunting surveillance to be useful is (1) a relative risk in the form of a clear infection pattern among classes of units, (2) selective harvesting is possible based on these units, and (3) a population model to predict one year ahead. In our case, we have a clear sex and age-specific pattern of CWD infection, and hunting of cervids is typically selective for these classes. Our model can easily be modified to explore a wider range of demographic infection patterns. In principle one may also use other factors leading to relative risks, either temporal (seasonal infection) or spatial (distance to known infection) depending on disease system details. "

L185 – provides

Our response: Typo corrected.

Table 1 – why is 'Principle and Basis' cell for 'Harvest rate and composition' blank?

Our response: It was since the harvest rate and composition is an output of our model, we have now made this explicit by adding "output from the model and dependent on harvest strategy: average historical harvest rates or based on operational sex ratio" in this box.

Fig. 2 caption – what do the different colored text mean?

Our response: As written in the legend, those in red are key parameters. We have now added what we mean by key parameters: "The model is run for various combinations of selected values of key parameters (marked in red) and compared with respect to the time to reach confidence in freedom from infection (cfr. Table 1)"

Fig. 3 caption – I'm not sure what the 'tested' trajectory is representing; what is the skew?

Our response: We have now changed 'tested' to 'emp. data' to make it clear this was based on the testing of animals that was harvested/culled. We now make it clear that the skew in sex ratio is the degree of female bias: "..... the principle of proactive hunting surveillance with a male biased harvest resulting in a female biased sex ratio in the remaining population ('emp. data')."

Fig. 4 – I don't find this figure very informative as it's completely predictable from the harvest strategies. It would be better to predict population abundance or growth rates as a metric of impact of the different strategies

Our response: We have followed this good advice and made a new figure that we hope is more informative.

SUPP info – L67 – it would be useful to provide some description of the type of model from Nilsen and Strand so one doesn't have to go looking, i.e., Bayesian hierarchical is super vague so the reader gets no insight as to which data are needed and how they are used.

Our response: We have added a more detailed description of the model:

“Based on the census data, we estimated reindeer abundance, population age- and sex-structures as well as demographic rates (i.e. survival probabilities and reproductive rates) using a hierarchical change-in-ratio model with parameters estimated using Bayesian inference described in Nilsen & Strand (2018). In this model, information about changes in population structures following a removal of a known number of reindeer with known age- and sex distribution by harvest was utilized to estimate the parameters of interest. Estimating parameters based on Bayesian inference allowed us to exploit the great flexibility of the BUGS language, facilitating the use of data from a sampling scheme that does not match the assumptions of traditional estimators. As a core in the state model, we used a stage-dependent two-sex matrix model with three age classes: calves, yearlings and adults”.

SUPP info – L80 – same as above. There is not enough info here to understand what kind of model this is and which data are needed to use it.

Our response: We have added a more detailed description of the model: “The main principle of the disease detection model is presented in Viljugrein et al. (2019). This scenario-tree model estimates the likelihood of detecting CWD-infection in a given individual depending on its age, knowledge about infection development, and sensitivity of the testing regime used. We here extend the previous model to account also for sex-specific pattern of infection. The expected duration of time from infection to death can vary in the model, but we used a constant of 2 years for this purpose. The disease detection model relies on an understanding of the ability of the ELISA method to detect abnormal prion protein (PrPSc) during the course of CWD infection from samples of retropharyngeal lymph nodes (RLN) and brain tissue provided by hunters. For simulations of hunting for the years to come, we assumed that all reindeer that were shot, were sampled and tested. Furthermore, that the proportion of samples containing both RLN and brain tissue were set equal to the last year of data (2018) from the specific areas. We used the model to estimate the annual surveillance sensitivity for two adjacent populations of Nordfjella zone 1, that is Nordfjella zone 2 and Hardangervidda (Figure 1). ”

Reviewers' Comments:

Reviewer #1:

Remarks to the Author:

I have read the response to reviewers comments and the updated manuscript. The authors have done a fine job responding to the comments and revising their manuscript accordingly. When the authors were unable to adjust their framework in response to a comment (e.g., Reviewer 2's main concern), the authors included text within their revision that identifies the caveat within their study. Otherwise, all comments were basically integrated into the revision.